# Novel viruses of the family *Partitiviridae* discovered in *Saccharomyces cerevisiae*

**Nathan T. Taggart[1], Angela M. Crabtree[1☯, Jack W. Creagh[1☯, Rodolfo Bizarria, Jr.[1,2,3],
Shunji Li[1], Ignacio de la Higuera[4], Jonathan E. Barnes[5], Mason A. Shipley[1], Josephine
M. Boyer[1], Kenneth M. Stedman[4], F. Marty Ytreberg[5,6], Paul A. Rowley[1]***

**1** Department of Biological Sciences, University of Idaho, Moscow, Idaho, United States of America,
**2** Department of General and Applied Biology, São Paulo State University (UNESP), Rio Claro, São Paulo,
Brazil, **3** Center for the Study of Social Insects, São Paulo State University (UNESP), Rio Claro, São Paulo,
Brazil, **4** Center for Life in Extreme Environments, Department of Biology, Portland State University, Portland,
Oregon, United States of America, **5** Institute for Modeling Collaboration and Innovation, University of Idaho,
Moscow, Idaho, United States of America, **6** Department of Physics, University of Idaho, Moscow, Idaho,
United States of America

☯ These authors contributed equally to this work.
* prowley@uidaho.edu

ppat.1011418

Canada, CANADA

**Data Availability Statement:** All relevant data are
within the manuscript and its Supporting
Information files.

## Abstract

It has been 49 years since the last discovery of a new virus family in the model yeast *Saccharomyces cerevisiae*. A large-scale screen to determine the diversity of double-stranded
RNA (dsRNA) viruses in *S. cerevisiae* has identified multiple novel viruses from the family
*Partitiviridae* that have been previously shown to infect plants, fungi, protozoans, and
insects. Most *S. cerevisiae* partitiviruses (ScPVs) are associated with strains of yeasts isolated from coffee and cacao beans. The presence of partitiviruses was confirmed by
sequencing the viral dsRNAs and purifying and visualizing isometric, non-enveloped viral
particles. ScPVs have a typical bipartite genome encoding an RNA-dependent RNA polymerase (RdRP) and a coat protein (CP). Phylogenetic analysis of ScPVs identified three
species of ScPV, which are most closely related to viruses of the genus *Cryspovirus* from
the mammalian pathogenic protozoan *Cryptosporidium parvum*. Molecular modeling of the
ScPV RdRP revealed a conserved tertiary structure and catalytic site organization when
compared to the RdRPs of the *Picornaviridae*. The ScPV CP is the smallest so far identified
in the *Partitiviridae* and has structural homology with the CP of other partitiviruses but likely
lacks a protrusion domain that is a conspicuous feature of other partitivirus particles. ScPVs
were stably maintained during laboratory growth and were successfully transferred to haploid progeny after sporulation, which provides future opportunities to study partitivirus-host
interactions using the powerful genetic tools available for the model organism *S. cerevisiae*.

## Author summary

This article describes the discovery and characterization of multiple strains and species of
viruses from the family *Partitiviridae* in the brewer's and baker's yeast *S. cerevisiae*. These
novel viruses have bipartite genomes packaged in spherical viral particles with structural

**Funding:** Research reported in this publication was supported by the National Institute of General Medical Sciences of the National Institutes of Health under award number P20 GM104420 (PAR, FMY), the National Science Foundation Division of Molecular and Cellular Biosciences, grant numbers 1818368 and 2025305 (PAR), and the National Science Foundation EPSCoR Research Infrastructure Improvement Program: Track-2, award number OIA-1736253 (FMY). Computer resources were provided in part by the Research Computing and Data Services of the Institute for Interdisciplinary Data Sciences, sponsored by the National Institutes of Health under award number P30 GM103324 (FMY). This research also used the computational resources provided by the high-performance computing center at Idaho National Laboratory, which is supported by the Office of Nuclear Energy of the U.S. DOE and the Nuclear Science User Facilities under Contract No. DE-AC07-05ID14517 (FMY). RB supported by Fundação de Amparo à Pesquisa do Estado de São Paulo (FAPESP) grant #2021/09980-4. Undergraduate researchers were funded by the University of Idaho Office of Undergraduate Research and the Department of Biological Sciences (MAS, JMB). Undergraduate research in the Rowley laboratory is also supported by a Dyess Faculty Fellowship, University of Idaho College of Science (PAR). The funders had no role in study design, data collection and analysis, decision to publish, or preparation of the manuscript.

**Competing interests:** The authors have declared that no competing interests exist.

homology to members of the family *Partitiviridae*. Strikingly, yeast partitiviruses are most closely related to viruses from human pathogenic protozoa and not partitiviruses of other fungi. As partitiviruses can positively and negatively contribute to a host's physiology (including important human and plant pathogens), the presence of partitiviruses in *S. cerevisiae* offers a unique opportunity to study the biology of these viruses in a well-developed model system.

## Introduction

*Saccharomyces cerevisiae* is a yeast widely used for commercial fermentation to create diverse foodstuffs, including bread, beer, wine, coffee, and chocolate. It is also an important model organism for studying fundamental principles of metabolism and eukaryotic cell biology, including host-virus interactions. Like many fungi, *S. cerevisiae* harbors different parasitic genetic elements, including RNA viruses, DNA plasmids, and retrotransposons. These RNA viruses have no known extracellular stage in their lifecycle and transmit vertically during cell division or horizontally by mating [1,2]. Infection of fungi by viruses (mycoviruses) often does not result in overt signs of disease, but there are examples of how mycoviruses can influence fungal physiology [3–9].

The discovery of the first mycovirus in *S. cerevisiae* was due to the production of antifungal toxins, a phenotype often dependent on double-stranded RNA (dsRNA) viruses from the family *Totiviridae* and toxin-encoding satellite dsRNAs [10–12]. Single-stranded RNA viruses of *S. cerevisiae* (from the family *Narnaviridae*) were discovered by the detection of dsRNA replication intermediates [13]. Contemporary RNA sequencing techniques have enabled the discovery of many more mycoviruses, either from individual strains of fungi or from environmental samples, including novel viruses from the *Totiviridae* and *Narnaviridae* virus families in yeasts of the Saccharomycotina subphylum [14–18]. Due to its short generation time, ease of genetic manipulation, and the large array of genetic tools and other resources, *S. cerevisiae* is an ideal model organism for studying viruses and other parasitic genetic elements. Such studies have revealed mechanistic details relating to virus replication, persistence, and inheritance, as well as the cellular mechanisms that suppress viral replication [1,2].

Partitiviruses (PVs) have been found within the Basidiomycota and Ascomycota fungi, as well as plants and protozoa [19]. PVs are also associated with invertebrates and can actively replicate in insect cells and tissues, with some phenotypic changes to infected insects [20–22]. PV genomes comprise of two linear dsRNA molecules that range in size between 1.4 and 2.4 kbp. Each genome segment is separately packaged into a viral particle (i.e. biparticulate) [23]. One genome segment encodes the RNA-dependent RNA polymerase (RdRP) responsible for genome replication and transcription of viral RNAs. The second segment encodes the coat protein (CP) that encapsidates the viral genome [24]. Assembled PV virions typically measure 25–40 nm in diameter and have isometric T = 1 symmetry constructed of 60 homodimeric subunits per unenveloped particle. Five PV particle structures have been obtained, and all contain a striking protrusion domain (P-domain) that projects from the surface of the viral shell domain (S-domain) and forms intermolecular interactions between CP monomers [25–29]. PV particles can encapsidate two copies of each genome segment, and RNA transcripts are thought to be extruded from the virions into the cytoplasm for translation [30]. There are currently five approved genera within the family *Partitiviridae*, named *Alphapartitivirus*, *Betapartitivirus*, *Gammapartitivirus*, *Deltapartitivirus*, and *Cryspovirus*, based on criteria such as genome length, host species, and the relatedness of the RdRP and CP genes ([19,31] and

International Committee on Taxonomy of Viruses (ICTV): https://ictv.global/taxonomy/). The new genera *Epsilonpartitivirus* and *Zetapartitivirus* were recently proposed based on the discovery of several novel viruses [32,33]. PVs that persistently infect fungi have been found within all PV genera except *Deltapartitivirus* and *Cryspovirus*, which infect plants and protozoa, respectively.

PV infection was long considered cryptic, with few consequences for the infected cell. However, in fungi, infection by PVs can result in a slower growth rate, reduced conidia formation, altered morphology, and the disruption of sexual reproduction [34–38]. This can lead to hypovirulence in fungal pathogens relevant to agriculture [39–41] but not in all cases [33,35]. Of relevance to human health, PV infection can cause hypervirulence in the pathogenic fungus *Talaromyces marneffei* [42] and has been suggested to increase the fecundity of the pathogenic protozoan *Cryptosporidium parvum* [43]. The discovery of PVs in *S. cerevisiae* provides a unique opportunity to leverage the powerful genetic tools of this model organism to better understand their replication and effect on host physiology.

## Results

### Double-stranded RNA partitiviruses identified in *S. cerevisiae*

To take an unbiased approach to understanding virus diversity in *S. cerevisiae*, dsRNAs were purified from a random selection of 520 strains from the 1,002 Yeast Genomes Project collection (S1 Table) [44]. These dsRNAs were analyzed by a short-read sequencing pipeline that identified 110 contigs with significant sequence similarity to known virus families (Fig 1A and S2 Table) [45,46]. Contigs with sequences similar to the *Totiviridae* were most frequently

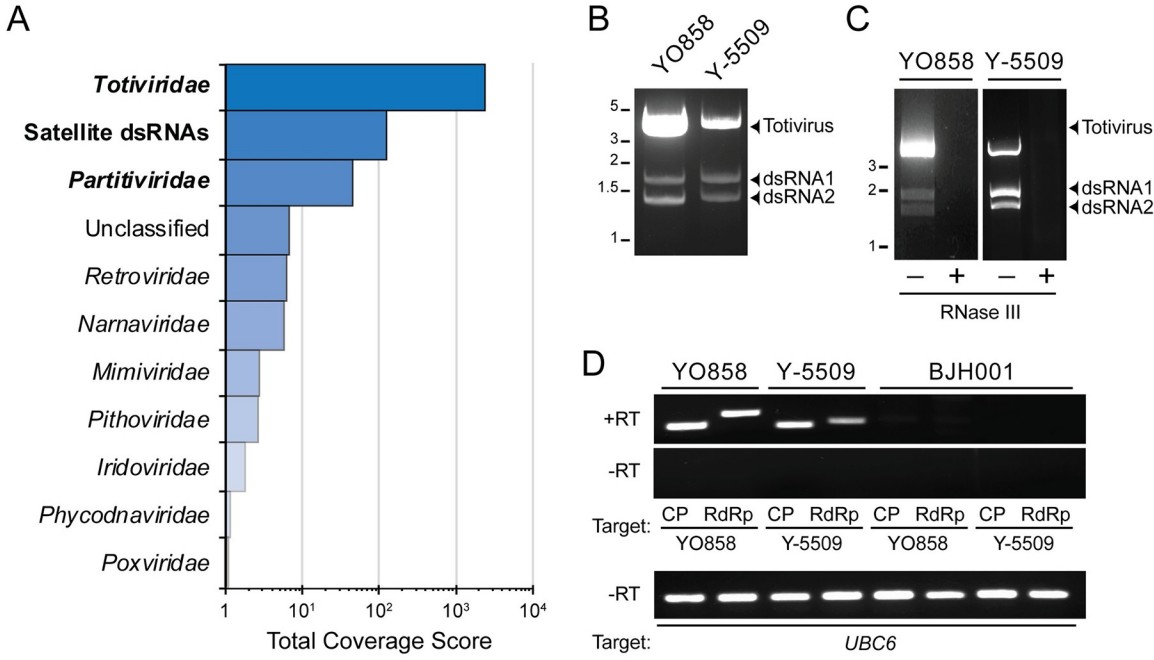

**Fig 1. The identification of PVs in *S. cerevisiae*.** (A) Virus contigs were assembled from the sequencing of dsRNAs extracted from 520 strains of *S. cerevisiae*. (B) Agarose gel electrophoresis analysis of dsRNAs extracted from two PV-infected strains of *S. cerevisiae*. (C) DsRNAs extracted in panel B were subjected to incubation at 37˚C with and without RNase III, and the resulting products were analyzed by agarose gel electrophoresis. (D) Two-step RT-PCR of total nucleic acids extracted from *S. cerevisiae* targeting the CP or RdRP genes with (+RT) or without (-RT) reverse transcriptase. Strain BJH001 was used as a negative control as it does not harbor PVs. Primers targeting the yeast gene *UBC6* were used as a PCR-positive control for genomic DNA.

identified. Contigs with similarity to satellite dsRNAs were also identified with ORFs corresponding to the *S. cerevisiae* killer toxins K1, K2, Klus, and *S. paradoxus* killer toxins K66 and K21. After totiviruses and satellites, the next highest coverage score was for viruses of the *Partitiviridae* [19]. Other virus families identified by these analyses were represented by three or fewer contigs and were not investigated further (S2 Table). Assembled contigs identified sequences similar to previously identified virus proteins in *Saccharomyces* yeasts that were not fully characterized [47,48].

To confirm the presence of PVs in *S. cerevisiae*, 161 strains from the 1,002 Yeast Genomes Project collection, 146 strains from the collection of Ludlow *et al.*, and three additional strains from other culture collections were screened using cellulose chromatography to extract dsRNAs (S3 Table) [44,49]. This approach identified PVs by the characteristic electrophoretic mobility of their dsRNAs, with 14% of strains containing PVs, mainly from the collection of Ludlow *et al* (Table 1). For example, purifying dsRNAs from *S. cerevisiae* strains YO858 and Y-5509 identified three distinct nucleic acid molecules with similar electrophoretic mobilities in both strains (Fig 1B). The RNAs were confirmed to be dsRNAs by RNase III digestion (Fig 1C). These dsRNAs likely represent a monopartite totivirus genome (≈4.6 kbp) and the bipartite genome of a PV (≈1.5 kbp) encoding an RdRP (dsRNA1) and CP (dsRNA2). Finally, reverse transcriptase-PCR (RT-PCR) confirmed PVs in total nucleic acid extracts of *S. cerevisiae* strains YO858 and Y-5509 (Fig 1D). PCR alone did not result in the amplification of PV sequences, and all samples were positive for the yeast chromosomal gene *UBC6* (Fig 1D). *S. cerevisiae* BJH001 was included as a negative control as it is not infected with PVs [45].

In *Saccharomyces* yeasts, dsRNAs smaller than 4 kbp have been associated with non-autonomously replicating satellites [1,2]. These satellites are often responsible for encoding antifungal killer toxins maintained by the replicative machinery of totiviruses. One strain of killer yeast (*S. cerevisiae* CYC1172 (Fig 2A)) was discovered to harbor dsRNAs corresponding to a totivirus and three species of low molecular weight dsRNAs (Fig 2B). This suggested the coexistence of a totivirus, PV, and a satellite dsRNA. To confirm that one of the three low molecular weight dsRNAs was a satellite, strain CYC1172 was grown on agar containing cycloheximide. This resulted in the curing of the satellite dsRNA and the loss of the killer phenotype, but not the curing of the totivirus or PV (Fig 2A and 2B). Moreover, screens of *S. cerevisiae* strains for dsRNAs identified 15 strains with PVs that lacked totiviruses (4.8% of all yeasts screened; Table 1), supporting the autonomy of PVs (Fig 2C).

## PVs from *S. cerevisiae* are most closely related to Cryptosporidium parvum virus 1

ScPVs were significantly enriched in strains isolated from coffee (25/44, Fisher's exact test p<0.01) and cacao (16/44, Fisher's exact test p<0.01) (Fig 3A) (S3 Table). Only three strains of

**Table 1. The prevalence of viruses in 310 strains of *S. cerevisiae*.**

| Type of dsRNA | No. strains | % prevalence |
|---|---|---|
| Totiviruses | 129 | 42 |
| Satellites | 63 | 20 |
| Partitiviruses | 44 | 14 |
| ScPV1* | 25 | 57 |
| ScPV2* | 32 | 73 |
| ScPV3* | 2 | 5 |

* Total number of ScPVs is higher than the number of *S. cerevisiae* strains due to virus coinfection.

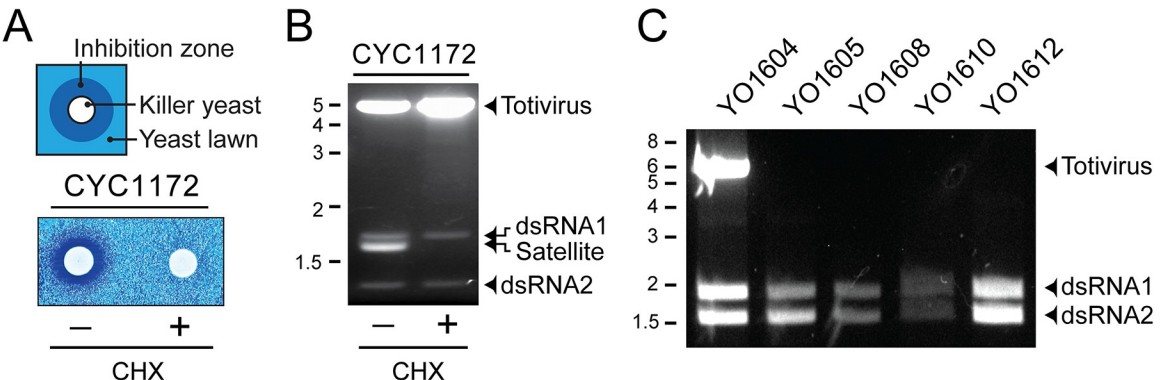

**Fig 2. The presence of ScPVs is independent of satellite dsRNAs and totiviruses.** (A) *Top* Schematic of a killer toxin production by a killer yeast strain with a zone of growth inhibition. *Bottom* Killer activity of strain CYC1172 before and after treatment with cycloheximide (B) Electrophoretic mobilities of dsRNAs extracted from CYC1172 before and after treatment with cycloheximide. (C) Agarose gel electrophoresis of dsRNAs extracted from selected strains of *S. cerevisiae* that contained either a totivirus and PV (lane 1) or only a PV (lanes 2–5).

*S. cerevisiae* with PVs (Y-5509, CYC1172, and ICV D254 Lalvin) were isolated from other environments (coconut pods and grape must). To characterize the genome sequence and organization of PVs from *S. cerevisiae* (ScPVs), complete ORFs were assembled for 14 ScPVs that corresponded to both an RdRP (486–491 amino acids) and a CP (280–310 amino acids) using short read sequencing (S1 Fig and S4 Table). ScPV proteins were smaller than PVs from plants, fungi, and protozoa, except for the RdRPs from the *Deltapartitivirus* genus (Fig 3B). An alignment of the RdRP from 14 ScPVs and 45 representative sequences from five official and two unofficial *Partitiviridae* genera was constructed for phylogenetic analysis using maximum likelihood. All ScPVs clustered as a single monophyletic group, with CSpV1 from the protozoan *C. parvum* as the most closely related outgroup (Fig 3C). The percent amino acid identity between ScPV and CSpV1 RdRPs was between 33–38%, and the CPs between 17–23% (S5 Table). The clustering of ScPVs suggests that there are three distinct species, which have been named *Saccharomyces cerevisiae partitivirus-1*, *-2*, and *-3*, represented by ScPVs isolated from strains Y-5509 (ScPV1-5509), YO858 (ScPV2-858), and CYC1172 (ScPV3-1172), respectively. The amino acid identities of the RdRP or CP from these three species are less than 90% identical, supporting their categorization as distinct species (S5 Table). The partial sequence of an ScPV (named ScCV1) was previously identified in a European oenological yeast strain [48], which has 98.78% amino acid sequence identity to ScPV3-1172, isolated from a European wine strain (La Rioja, Spain) in grape must.

Having identified many strains of *S. cerevisiae* with ScPVs, RT-PCR, and short-read sequencing were used to determine which species of ScPV were present in each yeast strain (S6 Table and Table 1). Eight strains harbored ScPV1, 16 strains ScPV2, and 16 strains were co-infected with ScPV1 and ScPV2. ScPV3 was only found in two European wine strains. Co-infection of strains with ScPV1 and ScPV2 was confirmed using gel electrophoresis by the resolution of doublets of dsRNA1 (S2 Fig).

The complete genome sequences of ScPV1-5509, ScPV2-858, and ScPV3-1172 were determined by 5' rapid amplification of cDNA ends (RACE) (Fig 4A). Overall, the bipartite organization of ScPVs was the same as CSpV1, with similar length genome segments and untranslated regions (UTRs). Analysis of the 5' UTRs of these viruses by mFold revealed commonalities in RNA secondary structure between genome segments of each ScPV and extensive structure in dsRNA2 (Fig 4A and S3 Fig). The sequence analysis of the 3' UTRs of ScPVs identified conserved sequence elements (CSEs) in dsRNA1 and dsRNA2 that were unique to each

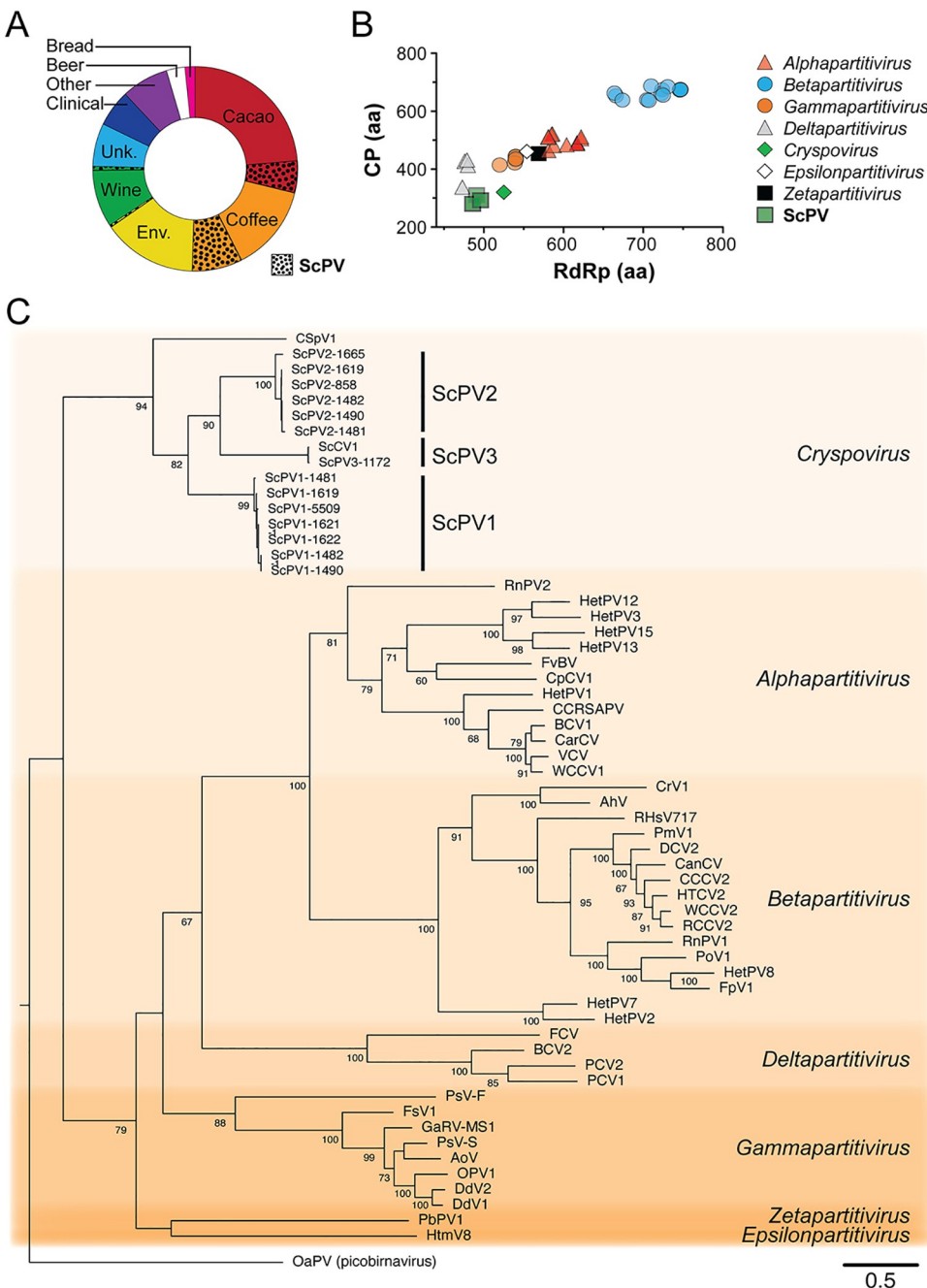

**Fig 3. ScPVs are closely related to *C. parvum* PVs** (A) Strains of *S. cerevisiae* isolated from different sources and the proportion infected with ScPVs (dotted regions). (B) Comparing the protein lengths of RdRPs and CPs of diverse PVs to ScPV1-5509, SCPV2-858, and ScPV3-1172. (C) A PhyML maximum likelihood phylogenetic model of the relatedness of PVs based on the amino acid sequence of the RdRP [50]. The RdRP from the picobirnavirus OaPV was used as an outgroup. The VT amino-acid exchange rate matrices were selected with a gamma distribution (G), a proportion of invariable sites (I), and empirical amino acid frequencies in the alignment fit these data best, as judged by PROTTEST [51]. The numbers at each node are the bootstrap values from 1000 iterations. The scale bar represents the distance of one amino acid substitution per site. The amino acid sequences in the phylogeny are from the viruses listed in S7 Table.

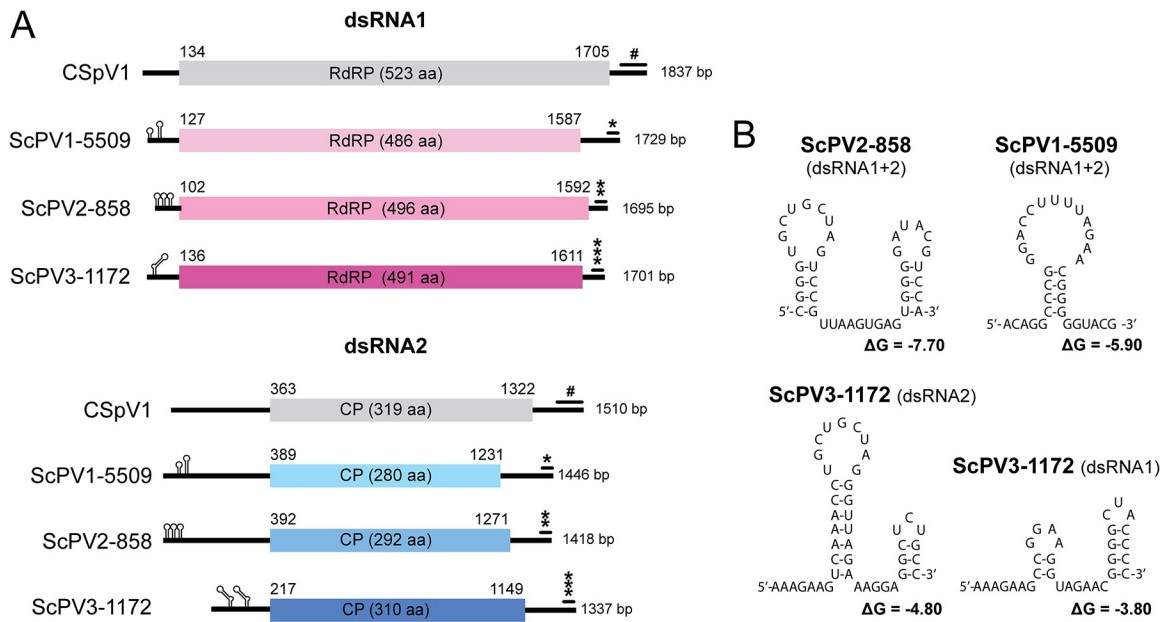

**Fig 4. Genome organization of PVs from *S. cerevisiae*.** (A) Schematic representation of dsRNA1 and dsRNA2 of three species of PV from *S. cerevisiae* as well as CSpV1. The ORFs are represented as rectangles that encode the RdRP and CP. Stem-loop structures are annotated to represent similar structures in the 5' UTRs of each species (S3 Fig). #/*/**/*** represent the pairs of CSE sequences in the terminal 3' UTRs (B) Secondary structure of RNA sequence present in the 3' CSE of dsRNA1 and dsRNA2 of ScPV1-5509, ScPV2-858, and ScPV3-1172.

virus species, including CSpV1 (Fig 4B) [52]. The CSEs of ScPV1-5509 and ScPV2-858 were 32 bp and 40 bp, and 100% identical when comparing the UTRs of dsRNA1 and dsRNA2. ScPV3-1172 encoded a CSE of 36 bp but with 84% identity. There was no homology between CSEs from different ScPV species, and all were predicted to form small stem-loop structures that could be involved in viral RNA replication or packaging.

## ScPVs assemble spherical particles with structural similarity to the PV shell domain

Previously described PVs formed spherical particles that package viral RNAs for replication and transcription. To determine whether ScPVs assemble similar particles, a strain of *S. cerevisiae* only infected with ScPVs was used for viral particle purification. Spherical virus-like particles with a diameter of ≈30 nm that lacked obvious P-domain surface protrusions were observed by TEM after sucrose gradient fractionation (Fig 5A).

To determine the tertiary structure of the CP monomer of ScPVs, *ab initio* models were generated using AlphaFold2. The N-terminus of all ScPV CPs and the *Cryspovirus* CSpV1 were modeled with confidence of up to 80% that sharply declined after ≈150 amino acids (S4 Fig). Energy minimization improved the models resulting in an average molprobity score of 2.2 Å and an average of 81.7% Ramachandran favored residues (S5 Fig and S8 Table). The N-terminal domain of the ScPV CP was mostly α-helical and with a long α3-helix cradled by four shorter α-helices, (α2, α4, α5, and α6) forming a rhomboid shell domain (S-domain) prototypical of PVs (Fig 5B). DALI analysis of the ScPV CP revealed structural similarity to the CP of pepper cryptic virus 1 (PCV1; PDB:7ncr) with an average root mean squared deviation (RMSD) of 4.34 Å (Fig 5C). Superimposition of the ScPV CP in the context of a virus particle from PCV1 showed that the ScPV S-domains likely form the majority of the particle structure

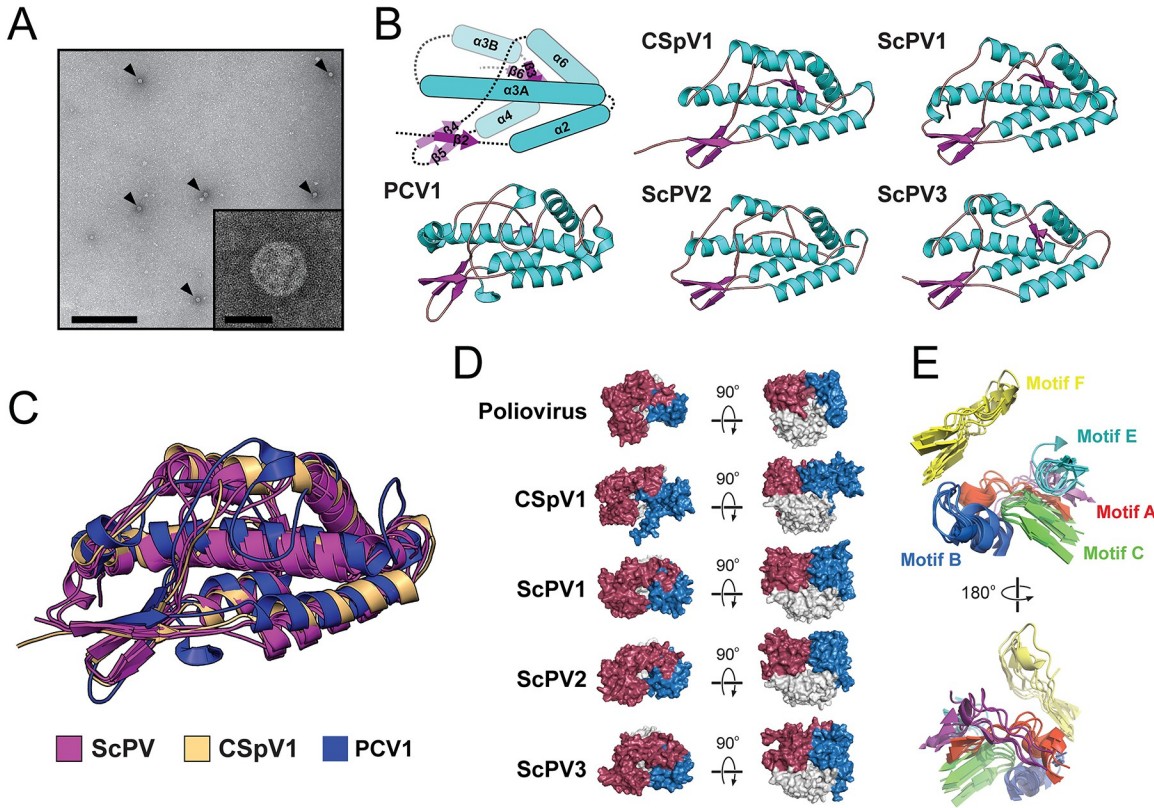

**Fig 5. The ScPV CP and RdRP share structural similarities with PVs and poliovirus.** (A) Visualization of ScPV1 particles by TEM as indicated by arrowheads. Scale bars are 500 nm (main image) and 25 nm (inset). (B) Molecular modeling using AlphaFold2 and energy minimization of the S-domain of three species of ScPV and CSpV1 compared to the solved structure of PCV1 (PDB:7ncr). The α-helices (blue rectangles) and β-sheets (magenta arrows) are labeled according to their common positioning within the S-domains. (C) An overlay of the five structures represented in panel B. (D) Molecular models of CSpV1, ScPV1, ScPV2, and ScPV3 RdRPs generated by comparison to poliovirus 1 RdRP (PDB:1ra6). Red, fingers domain; Blue, thumb domain; White, palm domain. (E) A superimposition of the RdRP conserved catalytic motifs A-F of poliovirus 1, CSpV1, ScPV1, ScPV2, and ScPV3. Red, motif A; Blue, motif B; Green, motif C; Purple, motif D; Cyan, motif E; Yellow, motif F.

(S6 Fig). Due to the lack of a confident modeling prediction for the C-terminal domain, it was unclear whether ScPVs encode P-domains.

## The RdRPs of ScPVs share structural homology with the polymerases of the *Picornaviridae*

The same molecular modeling approach as performed with the ScPV CPs was also applied to predict the tertiary structure of the RdRPs of ScPVs and CSpV1, with all models having 95% confidence across the entire sequence (S4 Fig). Energy minimization resulted in an average 93.5% Ramachandran favored residues and an average molprobity of 1.20 Å (S5 Fig and S8 Table). The tertiary structure organization of the modeled structures included the characteristic fingers, palm, and thumb domains, with a tertiary structure similar to human poliovirus RdRP (PDB:2ijd-1 and PDB:1ra6, average RMSD <3.1 Å) (Fig 5D). As expected, motifs A-F aligned well between ScPVs, CSpV1, and poliovirus (Fig 5E and S7 Fig). The close similarity of these RdRPs supports the common ancestry of the *Partitiviridae* and the *Picornaviridae* and their membership within the phylum *Pisuviricota*.

### ScPVs are maintained in *S. cerevisiae* under laboratory conditions and inherited in a non-mendelian manner after sporulation

To leverage the power of *S. cerevisiae* as a model system to study PVs, it was important to determine whether these viruses are stably maintained under laboratory conditions. Four strains of *S. cerevisiae* containing ScPVs were maintained on agar plates incubated at either room temperature or 4˚C for 6 weeks. The presence of ScPVs was monitored by the extraction of dsRNAs and indicated that ScPVs were stably maintained (S8 Fig). PVs have been shown to transmit inefficiently during spore formation in certain fungal species [31], which would limit genetic approaches to studying virus-host interactions in *S. cerevisiae*. To determine whether ScPVs can segregate efficiently to daughter cells during meiosis, sporulation was induced in two diploid strains infected with ScPVs. Total nucleic acids were extracted from each haploid progeny that were assayed for the presence of ScPVs by RT-PCR and by extracting dsRNAs (S8 Fig). ScPVs were identified in all four meiotic progenies. As ScPVs segregated in a non-Mendelian fashion, virus-containing strains were mated with the laboratory strain *S. cerevisiae* BY4741 to create ScPV-infected hybrid strains with the expected genotype (S9 Fig).

### ScPVs are resistant to the antiviral effects of Xrn1p

Prior studies have shown that pathways of RNA metabolism in yeasts are important for controlling the replication of totiviruses [9,53–58]. Specifically, the 5' exoribonuclease Xrn1p cures dsRNA totiviruses when overexpressed, likely because of the degradation of uncapped viral RNAs. To determine if ScPVs are susceptible to attack by Xrn1p, a high copy plasmid encoding *XRN1* was used to transform a strain of *S. cerevisiae* harboring totiviruses (L-A-lus and L-BC), an M2 satellite dsRNA, and ScPV3. After transformation, colonies were assayed for the loss of killer toxin production as an indication of Xrn1p restriction, as described previously [56]. The expression of *XRN1* led to 50% of clones losing K2 killer toxin expression (S10 Fig), which was comparable to the loss of K1 expression (57%) [56]. Extraction of dsRNAs from non-killer yeasts revealed the expected loss of M2 in all strains, whereas killer yeast always maintained M2. Conversely, dsRNAs from ScPV3 and totiviruses were still observable, even after the loss of M2 (S10 Fig). This demonstrated that ScPV3 appears resistant to the overexpression of *XRN1*, which is effective at curing the L-A totivirus [56,57].

## Discussion

This study reports the first detailed characterization of novel dsRNA viruses in *S. cerevisiae* in almost 50 years [12]. These viruses are from the family *Partitiviridae*, the first viruses of this type to be identified in yeasts of the Saccharomycotina subphylum [47,48]. ScPVs were predominantly associated with strains isolated from coffee and cacao beans from four genetically distinct yeast populations (South American and African cacao, South American and African coffee) that show evidence of admixture due to human migrations and farming practices [49]. ScPV1 and ScPV2 species were repeatedly found in yeasts isolated from coffee and cacao beans in different geographic locations, suggesting that human agricultural practices have distributed them with their host yeast strains [49]. ScPV3 is unique because it has only been identified in European *S. cerevisiae* strains associated with winemaking, which suggests either an independent lineage of ScPVs or outcrossing and invasion from strains associated with coffee and cacao.

Previous reports show that mycoviruses can be found in diverse species of fungi [59]. In *S. cerevisiae*, up to 22% of strains are thought to harbor mycoviruses due to the known association of viral infection with killer toxin production [46,60]. However, in coffee and cacao-

associated strains, 46% of strains harbor at least one dsRNA virus. Fermentation of coffee and cacao beans by different yeast and bacterial species, including genetically distinct strains of *S. cerevisiae*, is used to digest carbohydrates and proteins of the bean mucilage [61,62]. This could provide a unique environment for viral invasion of different yeast populations due to more frequent outcrossing and admixture. The prevalence of ScPVs in coffee and cacao strains could also be because they provide a specific advantage to *S. cerevisiae* during fermentation. As commercially available beans are the product of successful fermentation and have desirable flavor profiles, it would be interesting to survey failed fermentations to determine whether ScPVs are beneficial to coffee and cacao fermentations. Alternatively, it is also possible that coffee and cacao strains could be more susceptible to viral infection when compared to other yeast lineages, as genetic differences between strains and species of yeasts can render a strain more permissive to the replication of viruses and parasitic plasmids [9,56,58,63–66].

ScPVs share a common ancestry with PVs of the *Cryspovirus* genus isolated from the protozoan pathogen *C. parvum* and other species of *Cryptosporidium* [67,68]. ScPVs have diverged from protozoan viruses as evinced by the low amino acid sequence identity of their RdRP (33–38%) and CP proteins (16–23%). These identities are close to the proposed cutoffs that would define ScPVs as a novel virus genus. Moreover, the smaller sizes of ScPV proteins set them apart from all previously described PVs. The genus name *Cryspovirus* was originally proposed to distinguish protozoan PVs from plant PVs [69]. The discovery of the ScPVs could justify reassessing the current taxonomic organization of the *Partitiviridae*.

The evolutionary relationship between ScPVs and CSpV1 suggests horizontal PV transfer between yeasts and protozoa. *Cryptosporidium spp.* have likely been associated with humans and livestock for millennia [70], and both can be chronically infected, shedding thousands of infectious oocysts per gram of feces. Similarly, yeasts are propagated in high numbers during fermentation. Together, this would release significant numbers of PV-containing cells into the environment that could have increased the likelihood of horizontal virus transfer. Although PVs lack genes that would enable mobilization, the extracellular transmission of viral particles has been demonstrated under laboratory conditions for both mycoviruses and protozoan viruses [71–73]. Previously, cross-kingdom virus transmission between fungi and plants has been shown to be possible, as inferred by phylogenetic discordance between virus and host, and has also been observed in the laboratory [74–76]. Additional efforts to characterize fungal and protozoan viruses will be vital in understanding the evolutionary origins of PVs and the frequency of viral transmission between kingdoms.

The CPs encoded by ScPVs are the smallest described in any member of the *Partitiviridae* with significant structural homology to other PV CPs, despite a lack of sequence homology. In other CP structures, the S-domain is larger because of extended loops between helices and the addition of P-domains that can interrupt the S-domain or are added as C-terminal extensions [26–29]. The predicted structure of ScPV CPs appears to most closely resemble the structure of the PCV1 CP but lacks an obvious P-domain in the S-domain [25]. The C-terminal domain of ScPV CP is 100 amino acids long and would be sufficient to form a P-domain, as they are ≈80 amino acids in length. The observed smooth particle morphology of ScPV1 would disfavor a prominent P-domain as seen with PCV1 and might be more similar to the "butte" domain of Fusarium poae virus 1 [29]. The ScPV C-terminus could play a role in virus particle assembly as in other PVs, but further empirical analysis will be needed to determine its role in particle assembly and stability.

The detailed study of host-virus interactions has been limited for PVs due to their infection of non-model organisms with a paucity of genetic tools. The similarity between ScPVs and CSpV1 would also enable a better understanding of PVs that infect a pathogenic protozoan without the need to collect oocysts from feces or the use of tissue culture models of *C. parvum*

infection. *S. cerevisiae* has been used for decades to study the replication of totiviruses and narnaviruses by using well-developed techniques that allow genetic crosses, genome manipulation, and high-throughput screening. This has included developing a reverse genetic system for launching narnaviruses from cloned cDNAs, the characterization of the antiviral systems of yeasts, and the ectopic expression of the totivirus Gag-Pol [9,53–58,77,78]. However, the study of mycoviruses using *S. cerevisiae* as a model has been limited to those viruses that naturally infect *S. cerevisiae*, despite the successful laboratory infection with mycoviruses from filamentous fungi [79,80]. Therefore, the identification of ScPVs in *S. cerevisiae* and the discovery that they are transmitted during meiosis will allow the use of a myriad of genetic tools and strain collections to study host-virus interactions, antiviral mechanisms, and PV replication. Currently, it is unknown which cellular processes are required for efficient PV replication or how previously identified antiviral systems that restrict virus replication in yeasts interact with ScPVs [9,53–58]. ScPVs appear resistant to Xrn1p, which suggests that these viruses employ mechanisms to protect their RNAs from degradation. This could include 5' end modification, cap-snatching, or secondary structure to block Xrn1p progression, as has been shown for other *S. cerevisiae* viruses [55,57,81–83]. *S. cerevisiae* represents a unique opportunity to build on prior studies and further our understanding of the important and diverse members of the *Partitiviridae*.

## Materials and methods

### Double-stranded RNA extraction

Double-stranded RNA was purified according to the rapid method of Crabtree *et al.* [45]. The yield was increased by recovering the supernatant from multiple phenol-chloroform extractions and loading it onto a single cellulose column. All strain information and their dsRNA content are compiled in S3 Table.

### Sequencing of dsRNAs

Samples were prepared for sequencing by adding 3' poly(A) tails to purified dsRNAs and used as priming sites for cDNA synthesis, as previously described [45,46]. DNAs were prepared for Illumina sequencing using tagmentation [84], and sequencing of cDNAs was carried out according to Crabtree *et al.* [45]. For determining the sequence of the ends of PV genomes, the 5' RACE kit was used from Invitrogen as per the manufacturer's instructions. Completing the ScPV1 RdRP ORF required the following modifications to the manufacturer's protocol: Superscript IV reverse transcriptase (Invitrogen) was used instead of Superscript II. A longer GSP1 was designed to suit the higher temperatures of cDNA synthesis. And the 37°C TdT incubation time was decreased to 2.5 min. Primers used to amplify the 5' ends of the segments of the various PVs are listed in S9 Table. 5' RACE products were cloned into the pCR8 vector using the TOPO-TA cloning kit (Thermo Fisher Scientific). All complete genome sequences of ScPVs have been deposited to NCBI Genbank under the accession numbers OP555746 (ScPV2-858 dsRNA2; CP), OP555747 (ScPV2-858 dsRNA1; RdRp), OP555748 (ScPV1-5509 dsRNA1; RdRP), OP555749 (ScPV1-5509 dsRNA2; CP), OP555750 (ScPV3-1172 dsRNA1; RdRP), and OP555751 (ScPV3-1172 dsRNA2; CP). All ScPV sequence accession numbers are listed in S10 Table.

### Sequencing analysis of dsRNA metagenomic data from *S. cerevisiae*

109 contigs (>300 nt in length) were identified as matching "Viruses (taxid: 10239)" using BLASTx (Max targets 10, Threshold $1 \times 10^{-6}$). Each hit was classified by virus family, and the coverage scores were totaled.

### RNAse III digestions

Double-stranded RNAs were digested with ShortCut RNase III (New England Biolabs, Ipswich, MA, USA) according to the manufacturer's protocol.

### Total nucleic acid extraction from *S. cerevisiae*

Total nucleic acids were extracted from saturated liquid cultures of yeasts for RT-PCR as described previously [85].

### Reverse transcription-PCR

Superscript IV (Invitrogen) was used for all RT-PCR reactions as directed by the manufacturer. Primers used to detect PVs are listed in S8 Table.

### Killer phenotype assays

Killer toxin production by yeasts was measured using killer yeast agar plates (yeast extract, peptone, and dextrose (YPD) agar plates with 0.003% w/v methylene blue buffered at pH 4.6), as described previously [46].

### Cycloheximide curing of satellite dsRNAs

10 μL of an overnight YPD culture of *S. cerevisiae* strain CYC1172 was transferred to YPD plates containing up to 1.5 mg/L of cycloheximide and incubated at ambient temperature for 3–5 days. Colonies growing on the highest concentration of cycloheximide were streaked to single colonies on standard YPD media and incubated at ambient temperature for 3–5 days. Ten colonies were assayed for the loss of the killer phenotype as described under "killer phenotype assays."

### Phylogenetic analysis

RdRP protein sequences from ScPVs and other PVs from previously described genera (S7 Table) were aligned using MUSCLE and were inspected manually for accuracy. PROTTEST3 v3.4.2 was used to determine each dataset's appropriate amino acid substitution matrix [51]. The search space used consisted of all possible matrices and decorations with eight categories for the +I and +I+G models. The best-fit model was determined to be VT+I+G+F, with the next best model having a delta AIC of 28.52. PhyML v3.3.20220408 was used to create phylogenetic models [50]. PhyML parameters were chosen based on the results of PROTTEST3. The analysis consisted of 1000 bootstrap replicates, the VT amino acid substitution model was used, the proportion of invariable sites was determined by maximum likelihood, amino acid frequencies were estimated empirically by the frequency of occurrence in the dataset, the BEST tree search method was used for tree estimation, and the analysis was started with five random trees.

### Virus particle purification

This protocol was adapted from that of Naitow *et al.* [86]. Cells were inoculated in 1000 mL yeast peptone dextrose (YPD) liquid media and incubated at 30˚C for 24 hours. Cells were harvested by centrifugation for 5 min at 3800 × g and washed with double-deionized water. Washed cells were suspended in spheroplasting buffer (100 mM Tris-HCl, 1M D-sorbitol, 20 mM β-mercaptoethanol, 0.5 mg/mL Zymolyase 20T, pH 7.6) and incubated for 90 min at ambient temperature, stirring gently. Digested cells were harvested by centrifugation for 5 min

at 3,800 × g and suspended in buffer A (50 mM Tris-HCl, 150 mM NaCl, 5 mM EDTA, 1 mM dithiothreitol, pH 7.6), after which they were lysed by passing the cell suspension twice through a French pressure cell press (Thermo Scientific) at 20,000 psi. Cell debris was removed by centrifugation for 20 min at 9,600 × g, and the supernatant was adjusted to 3% PEG-8000 and 0.5 M NaCl before rocking gently for 60 min at 4˚C. The precipitate was harvested by centrifugation for 20 min at 9,600 × g, gently suspended in 15 mL of 50 mM sodium phosphate buffer (pH 7.0), and incubated on ice overnight with rocking. Insoluble precipitates were removed by centrifugation at 7,000 × g for 10 min. The supernatant was layered onto 5 mL 20% sucrose cushion in an ultracentrifuge tube (Thermo Scientific, Nalgene High-Speed Round-Bottom PPCO Centrifuge Tubes) and centrifuged at 80,000 × g for 2 hours at 4˚C. The pellet was recovered and suspended in 500 μL sodium phosphate buffer and layered onto a discontinuous sucrose gradient (10–50% in 5% increments of 1 mL each). This was centrifuged at 70,000 × g for 2 hours at 4˚C.

### Transmission electron microscopy

For the negative staining of VLPs, 5 μl of purified particles from strain YO1126 were applied to a 400-mesh copper grid coated with carbon-Formvar (Ted Pella) for 20 seconds, wicking with a filter paper, and washed three times on 50 μl water droplets. A staining solution of 2% uranyl acetate (pH ≈ 3) was added to the sample-side of the grid, wicking, and reapplied for 20 seconds before removal by wicking and lateral aspiration. Images were obtained on an FEI Tecnai F20 transmission electron microscope, captured with a BM UltraScan camera, and stored in digital micrograph 3 format.

### Molecular modeling

Collabfold version 1.4 was used to generate models using the open reading frame of the CP and RdRP of viruses CSpV1, ScPV1-5509, ScPV2-858, ScPV3-1172 with the following parameters (msa_mode: MMseqs2 (uniref+environment); model_type: auto; pair_mode: unpaired+paired; num_recyles: 3). Five unrelaxed structures of each CP and RdRP were generated. RdRP models were generated with similar local distance difference test (pLDDT) values per residue (S4 Fig). The highest average pLDDT score for each model was chosen for energy minimization and analysis. Energy minimization was carried out for each model using standard energy minimization protocol described in our previous study [17]. A dodecahedron box was generated around each model and solvated using a 10 Å layer of TIP3P. Ions were added to maintain charge neutrality at a concentration of 0.15 mol/L using $Na^+$ and $Cl^-$ ions. The force field parameter used for protein and ions was AMBER99SB*-ILDNP. Each system underwent energy minimization using the steepest descent algorithm for 10,000 steps via the GROMACS package. Stereochemical analysis was carried out with the SWISS-MODEL structure assessment tool for each model after the energy minimization (https://swissmodel.expasy.org/) to verify that model sterics were physically reasonable after energy minimization. All PDB formatted files for each relaxed model can be found in S1 File. The final energy minimized RdRP and CP model structures were submitted to the DALI server to identify structural homologs (DALI data presented in S2 File). The cealign command in the PyMOL visualization software package was used to identify regions of homology between CPs and homologous CP structures identified by DALI [87].

### Yeast tetrad dissection

*S. cerevisiae* cells were grown overnight at 30˚C in 2 mL YPD broth. 250 μL were added to 2 mL spo media (0.25% yeast extract, 0.25% dextrose, 1.5% potassium acetate, 0.002% (histidine,

leucine, lysine, tryptophan, methionine, arginine), 0.004% (adenine, uracil, tyrosine), 0.01% phenylalanine, 0.035% threonine, filter-sterilized). The sporulation rate was monitored by microscopy. When the sporulation rate was at least 25%, 50 μL of cell suspension was centrifuged, and the supernatant was removed. Cells were suspended in 50 μL 1 mg/mL yeast lytic enzyme (A. G. Scientific) and digested for precisely 12 min, after which the reaction was quenched by adding 0.5–1 mL sterile water. Asci were dissected with a 25 μm fiber optic cable glued to a micromanipulator mounted onto a Nikon Eclipse 50i microscope. Ascospores were germinated at 30˚C, grown to single colonies, and used for additional analysis.

## Transmission of ScPVs

Stable haploids of strains YO1126 and CYC1172 were obtained by *HO* disruption and sporulation to create the diploid strains YTag006 and Ytag015, respectively. Primers NTT033 and NTT034 were used to amplify the *ho*::*HygMX* cassette from strain YJM975 *MATα ho*::*HygMX*, using purified chromosomal DNA as a template. 1 μg of the amplicon was used to transform YO1126 and CYC1172. After selection on hygromycin (0.3 mg/mL), PCR verification was performed by probing *HO* with the same primers. *HO/ho*::*HygMX* diploid cells were sporulated and dissected. The dissection plate was replica-plated onto YPD containing hygromycin. Mating type was determined by PCR, using primers NTT041, NTT042, and NTT043. This procedure created the stable haploid strains YTag009 and YTag058 from YTag006 and Ytag015, respectively. YTag009 and YTag058 were each crossed with BY4741 using a method from the Meneghini laboratory (personal communication). Briefly, YTag009 was mixed with BY4741 in a ratio of ≈ 1:100 and spotted onto YPD agar and incubated at 30˚C overnight. Cells were streaked to single colonies on YPD plates on which had been spread 125 μL 1 mg/mL alpha factor pheromone and incubated at 30˚C for 1–2 days; this process was repeated once more. Diploids were verified by PCR, using primers NTT041, NTT042, and NTT043. This method created YTag121 and YTag085. ScPV infection was verified by RT-PCR and dsRNA extraction followed by agarose gel electrophoresis.

## ScPV stability

Yeast strains were streaked onto two YPD agar plates each and incubated at either 25˚C or 4˚C for 6 weeks. Cells were cultured in 12 mL YPD broth and incubated at 25˚C for 2 days, and their dsRNAs were extracted and analyzed by agarose gel electrophoresis.

## Halo mating type assays

Strains DBY7730 and DBY7442 were streaked onto YPD agar and incubated at 30˚C for 1–2 days. One colony from each was used to inoculate overnight cultures in YPD broth, shaken at 30˚C. The cell suspensions were diluted 1:20, and 400 μL of each was spread onto a 100 mm YPD agar plate and incubated at 30˚C for 30 min. Strains of interest were pinned onto the lawn using sterile toothpicks, and the plates were incubated overnight at 30˚C. A zone of growth inhibition surrounding a colony indicated that it is the opposite mating type of the lawn strain.

## Curing of viruses and dsRNA satellites from *S. cerevisiae* by *XRN1* expression

The strain *S. cerevisiae* YTag085 (an M2 killer yeast) was transformed with the high copy plasmid pPAR219 to express *XRN1* cloned from *S. cerevisiae* [56]. The loss of the killer phenotype in 183 clones was confirmed by a killer assay using two different K2-susceptible strains of yeast

(BY4741 and DBY7730). Cured clones were then assayed for dsRNAs by cellulose chromatography.

## Strains of yeast

All strains of yeast used in this manuscript are listed in S11 Table.

## Supporting information

**S1 Fig. Contig size and coverage plots from the sequencing of dsRNAs from 10 strains of *S. cerevisiae* with PVs.** Each point represents a single contig generated by RNA sequencing. Red points are contigs that had sequence homology to known PVs by BLASTx.
(TIF)

**S2 Fig. Gel electrophoresis of dsRNAs isolated from strains of *S. cerevisiae* that are co-infected with ScPVs.** To corroborate the results of the multiplex RT-PCR screen, dsRNAs from nine strains were extracted and electrophoresed in a 3.2% agarose gel slab at 140V for 245 min and stained with 0.5 μg/mL ethidium bromide for 40 min.
(TIF)

**S3 Fig. RNA secondary structure predictions of the 5' UTRs of ScPVs using mFold.** Numerals are used to denote similar stem-loop structures in the RNA, and arrows are used to mark bulges.
(TIF)

**S4 Fig. Local difference distance test per residue (pLDDT) output from alphafold2 runs of each CP and RdRP model.**
(TIF)

**S5 Fig. Ramachandran plots of relaxed and unrelaxed AlphaFold2 models of the CP and RdRP of ScPVs and CSpV1.** Ramachandran plots of AlphaFold2 models for CP and RdRP proteins. Each point represents an amino acid residue and its ϕ and ψ bond angles. The shaded area of the plots represents standard angles for α-helices, β-sheets, and left-handed helices from a database of 12,521 non-redundant experimental structures.
(TIF)

**S6 Fig. Secondary structure comparison of CP proteins from different PV.** (A) Secondary structures were derived from crystal structures of the CP of PCV1 (PDB:7ncr) and PsV-F (PDB:3es5) and molecular models of ScPVs and CsPV1. The α-helices and β-sheets are represented as blue and magenta boxes, respectively. The P-domain α-helices and β-sheets are numbered relative to PCV1, with the commonalities between structures highlighted in gray. Unstructured polypeptide chains are represented in orange. The C-terminal domains of ScPVs and CsPV1 that were not modeled with high confidence are represented by dashed lines. (B) A comparison between the native PCV1 particle structure and replacing a CP monomer from PCV1 (left panel, beige) with a single CP monomer from ScPV1 (right panel, light blue).
(TIF)

**S7 Fig. The RdRPs of ScPVs have motifs conserved with RdRPs from CSpV1 and poliovirus.** (A) Domain diagram of the RdRP of ScPV2-858 showing the position of motifs A-F. (B) A multiple sequence alignment of the residues of the RdRP conserved catalytic motifs. Red text indicates residues 100% conserved between the positive sense RNA viruses [88], CSpV1, ScPV1, ScPV2, and ScPV3.
(TIF)

**S8 Fig. ScPVs segregate into meiotic progeny and are stably maintained under laboratory conditions.** (A) A micrograph depicting *S. cerevisiae* asci. (B) Schematic of non-Mendelian inheritance of ScPVs during sporulation. Detection of ScPV using RT-PCR in the haploid progeny of four dissected asci of (C) Y-5509 and (D) *bottom panel* CYC1172. (D) Detection of ScPV3 in the haploid progeny of a dissected asci of CYC1172 by extracting dsRNAs. (E) The presence of ScPV was measured by dsRNA extraction after maintenance on agar plates incubated at either room temperature or 4°C for 6 weeks.
(TIF)

**S9 Fig. Creation of a hybrid strain of *S. cerevisiae* with ScPVs.** (A) Confirmation of expected genotypes using selective growth media. (B) Confirmation of the inheritance of ScPVs during strain construction by cellulose chromatography and gel electrophoresis.
(TIF)

**S10 Fig. ScPVs are insensitive to the expression of *XRN1*.** (A) Curing of the M2 satellite dsRNA from YTag085 by *XRN1* expression as assayed by an agar plate killer assay (B) Extraction of dsRNAs from killer yeasts isolated from the plate in panel A. CYC1172 and YTag063 (killer yeast strains) and BY4741 (non-killer yeast strain) were included as controls for the presence or absence of a satellite dsRNA. (C) Loss of M2 and presence of ScPV3-1172 from representative non-killer strains isolated from the plate in panel A.
(TIF)

**S1 File. PDB's files for all models (zip file).**
(ZIP)

**S2 File. DALI server data (zip file).**
(ZIP)

**S1 Table. List of strains from the 1002 Yeast Genomes Project analyzed for dsRNAs by next generation sequencing.**
(TSV)

**S2 Table. BLASTx analysis of contigs identified after sequencing dsRNAs from 520 strains of *S. cerevisiae*.**
(CSV)

**S3 Table. Description of strains surveyed for the presence of ScPVs by dsRNA extraction.**
(TSV)

**S4 Table. Names, origin, and ORF length of representative strains of ScPV.**
(TSV)

**S5 Table. The percentage identity between the CP and RdRP proteins of ScPVs and CsPV1.**
(TSV)

**S6 Table. The detection of ScPVs in strains of yeasts known to harbor dsRNAs using RT-PCR or short read sequencing.**
(TSV)

**S7 Table. Names, abbreviations, and accession numbers of PVs used in the phylogenetic analysis of ScPVs.**
(TSV)

**S8 Table. Molecular modeling structure assessment scores.**
(TSV)

**S9 Table. Primers used in this study.**
(TSV)

**S10 Table. Accession number for all ScPV sequences determined in this study.**
(CSV)

**S11 Table. Yeast strains used in this study.**
(TSV)

## Acknowledgments

The authors would like to acknowledge the Dudley laboratory (Pacific Northwest Research Institute) for useful discussions and for providing the *S. cerevisiae* strains isolated from coffee and cacao beans. We also want to thank the ARS Culture Collection (NRRL), Complutense Yeast Collection, and the Fungal Genome Stock Center for all other yeast strains. We also thank Dr. Jill Johnson, Dr. Marc Meneghini, Dr. Maitreya Dunham, and Emily Mitchell for training and advice and the gift of yeast strains.

## Author Contributions

**Conceptualization:** Paul A. Rowley.

**Data curation:** Nathan T. Taggart, Angela M. Crabtree, Jack W. Creagh.

**Formal analysis:** Nathan T. Taggart, Angela M. Crabtree, Jack W. Creagh, Jonathan E. Barnes, Paul A. Rowley.

**Funding acquisition:** F. Marty Ytreberg, Paul A. Rowley.

**Investigation:** Nathan T. Taggart, Angela M. Crabtree, Jack W. Creagh, Rodolfo Bizarria, Jr., Shunji Li, Ignacio de la Higuera, Jonathan E. Barnes, Mason A. Shipley, Josephine M. Boyer, Paul A. Rowley.

**Methodology:** Nathan T. Taggart, Angela M. Crabtree, Jack W. Creagh, Rodolfo Bizarria, Jr., Shunji Li, Ignacio de la Higuera, Jonathan E. Barnes, Mason A. Shipley, Josephine M. Boyer.

**Project administration:** F. Marty Ytreberg, Paul A. Rowley.

**Supervision:** Kenneth M. Stedman, F. Marty Ytreberg, Paul A. Rowley.

**Visualization:** Jack W. Creagh, Ignacio de la Higuera, Paul A. Rowley.

**Writing – original draft:** Nathan T. Taggart, Paul A. Rowley.

**Writing – review & editing:** Nathan T. Taggart, Angela M. Crabtree, Jack W. Creagh, Rodolfo Bizarria, Jr., Kenneth M. Stedman, F. Marty Ytreberg, Paul A. Rowley.

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
