## [Decision Letter · Decision Letter 0]

14 Dec 2022

Dear Dr. Rowley,

Thank you very much for submitting your manuscript "Novel viruses of the family Partitiviridae discovered in Saccharomyces cerevisiae" for consideration at PLOS Pathogens. As with all papers reviewed by the journal, your manuscript was reviewed by members of the editorial board and by several independent reviewers. In light of the reviews (below this email), we would like to invite the resubmission of a significantly-revised version that takes into account the reviewers' comments.

We cannot make any decision about publication until we have seen the revised manuscript and your response to the reviewers' comments. Your revised manuscript is also likely to be sent to reviewers for further evaluation.

Sincerely,

Aiming Wang, Ph.D

Academic Editor

PLOS Pathogens

Shou-Wei Ding

Section Editor

PLOS Pathogens

Kasturi Haldar

Editor-in-Chief

PLOS Pathogens

orcid.org/0000-0001-5065-158X

Michael Malim

Editor-in-Chief

PLOS Pathogens

orcid.org/0000-0002-7699-2064

Reviewer's Responses to Questions

**Part I - Summary**

Reviewer #1: The paper no PPATHOGENS-D-22-01850 by Taggart and others reports on the virus hunting from 520 strains of yeast Saccharomyces cerevisiae collected worldwide and from diverse sources of origin. The authors took a next-generation sequencing approach with purified dsRNA fractions and identified many RNA viral contigs as well as a relatively small number of DNA virus contigs that appeared to be contaminants. The paper focuses on partitiviruses, termed Saccharomyces cerevisiae partitivurses (ScPVs), detected in most cases together with coinfecting Saccharomyces cerevisiae totivirus. To obtain singly infected yeast strains with ScPVs, over 300 additional yeast strains were screened for ScPVs. Three tentative partitiviruses, ScPV1 to ScPV3, were identified, which represent three new partitivirus species possibly in the genus Cryspovirus in the family Partitiviridae. Based on the phylogenetic relation between yeast ScPVs and protozoan cryspoviruses, the authors hypothesized that horizontal transfer between S. cerevisiae and protozoa might have occurred. It is of interest that while ScPV1 and ScPV2 were isolated form S. cerevisiae strains from coffee and cacao beans of different localities, ScPV3 was isolated from S. cerevisiae strains associated with winemaking in Europe. After confirming efficient vertical transmission of ScPVs to daughter cells, the authors transferred ScPV1 to ScPV3 to a laboratory strain BY4741 via mating. The authors claim this to be a foundation for future studies of ScPVs. Using electron microscopy and AlphaFold2 predictions, the authors suggested ScPVs to have a particle structure similar to those of other partitiviruses for which reconstructed 3D-particle structures are available.

Overall, the manuscript is well written in a succinct way and contains some new information as mentioned above. Particularly, the observation that yeast partitiviruses (ScPVs) are more similar to cryspoviruses than to alphapartitiviruses, betapartitiviruses or gammapartititviruses is interesting. Note that ScPVs are are hosted by the ascomycete S. cerevisiae and no cryspo-like viruses were reported from other filamentous ascomycetes from which many members of other genera, i.e., Alphapartitvirus, Betapartitivirus and Gammapartitivurs, are reported. However, this reviewer feels that the manuscript suffers from several major (paucity of novelty) and minor deficiencies and needs extensive revisions.

Reviewer #2: This study identifies a new family of dsRNA Partitiviruses that infect S. cerevisiae, perhaps the most widely studied and experimentally facile model organism. This is thus an interesting and important finding. Moreover, it is quite timely given recent invigorated attention given to viral diversity and antiviral systems in other organisms. Given their known and emerging roles in human health and agriculture, the mycovirus field in general seems understudied. This paper thus represents a significant contribution.

The quality of the writing and organization needs to be improved. Specifically, much of the detail in the Results section seems more appropriate for the Methods section; there are numerous places with very long paragraphs; the organization and narrative are confusing at parts. While there are a few experiments needed to improve the manuscript, most of the issues relate to these organization issues which will improve the readability. These will be discussed further in Part III.

**Part II – Major Issues: Key Experiments Required for Acceptance**

Reviewer #1: Major points:

1. It is not surprising that field-collected isolates of S. cerevisiae harbor partitiviruses, as similar virus hunting studies with filamentous ascomycetes have shown much greater diversity of RNA viruses (Myers et al. mBio 2020; Chiapello et al. Virus Evol 2020; Sutela et al. Virus Evol 2020; Ruiz-Padilla et al. mBio 2021). Importantly, Crucitti et al. (Viruses 2022) reported the genome sequences of ScPVs as different name, and this lowers the scientific impact of the current study. This reviewer is aware that ScPVs have been characterized more thoroughly in the current study, but the authors still need to do something like what is mentioned below.

As the authors mentioned in the Author Summary and Introduction, this study provides “a unique opportunity to study partitiviruses in a well-developed model system.” This reviewer would suggest that the authors show whether some S. cerevisiae mutants defective in antiviral pathways are more susceptible to ScPVs. Alternatively, unclassified RNA viruses (Fig. 1A) should be elaborated to broaden the scope of the manuscript.

2. Fig. 5A. Perhaps this reviewer missed it, but why no RT-PCR bands were detected in the middle sucrose gradient fractions that were expected to be rich in virus particles. According to the method used by the authors (page 20), this reviewer feels that virus particles should be enriched in the central portion of the sucrose gradient.

Reviewer #2: The analysis in figure 4 should be expanded and improved. Related to this, lines 225-233 are confusingly written. First the authors state that they deduced the full sequence of ScPV2-858. Then they compare it to an already complete genome for ScPV1-5509. Did they not also need to deduce this genome? Next, an incomplete comparison of the molecular features of the ScPV1 and ScPV2 genomes are presented. The authors had the right idea here, but need to do a much better job. They should present a comparison of representative genomes of ScPV1, ScPV2, ScPV3, and the closely related CSpV1. The 5' UTRs are unusually long compared with those of Totiviruses. Do they contain any sequences capable of forming stem loop structures? Might these be internal ribosome entry sites? The authors do not discuss the significance of the 3' stem loop structures. It seems likely they may serve as packaging signals? These are very important details that should be easy to attain. The authors need to address this.

In lines 323-324, the authors state that they have transferred these viruses to a commonly used laboratory strain (BY4741) through a cross and dissection. First, no results are shown fo this. Second, the language describing this is not very precise. Basically, the authors are introducing ScPVs into other strains through crossing. The resulting strain they have made is not BY4741, but a hybrid. To truly transfer the virus into BY4741, a cytoplasmic transfer experiment would need to be done. Short of this, serial backcrossing to BY4741 would be needed. This is not necessary, but at a minimum, some evidence of transmission of ScPVs to the BY4741 hybrid needs to be presented and the strains with their resulting genotypes at all score-able loci needs to be provided.

**Part III – Minor Issues: Editorial and Data Presentation Modifications**

Reviewer #1: Minor points:

The text contains scientific inaccuracy and careless errors.

Line 31 & 81. Should include insect (Cross et al., JVI 2020).

Line 32. “Increased fecundity of human” was really substantiated?

Lines 32-33. Confusing. If “plant pathogens” refer to altered virulence, it should be moved after “altered virulence of”.

L82. “4.8 kbp” is not correct. See Nibert et al. (Virus Res 2014).

L87. There are only five ICTV-approved partitivirus genera, and two additional ones were unofficially proposed by research groups.

L106. This reviewer wondered if this was really substantiated. This reviewer doubts it.

L135 & L225, L251, L268 and in other places. Should be “RT-PCR.” That is more commonly used.

L314. Italicize “Pisuviricota.”

L319. Not four?

L411. Partitivirus biology have been well explored (Vainio 2021, EoV 4th ed; Rose and Maiss 2021, EoV 4th edt). Inoculation methods are well-established for fungal partitiviruses (Sasaki et al., Arch Virol 2007). Cause-effect relationships are established for some fungal partitiviruses (Chiba et al., 2013; Virus Res, 2016). Should be more specific. See Vainio (2021, EoV 4th edt) and Rose and Maiss (2021, EoV 4th edt).

Fig. 1D. The “Totivirus” arrowhead mis-points, and should be lowered.

There are two Fig. S6s

Reviewer #2: It is difficult to follow the narrative for the varied yeast strains the authors queried. How is it that the authors went from the 520 strains to YO858, Y-5509, and CYC1172? Why did the authors choose to screen a different collection of yeast strains to elucidate the prevalence of ScPVs? Wouldn't it make more sense to just carry on with the original 520? Is it possible to consolidate these efforts?

As mentioned above, in several places, the paragraphs are very long (for example, line 159-179). The overall organization should be reconsidered.

Figures 1C and 1D are redundant.

The first sentence of the abstract is not precisely stated.

PLOS authors have the option to publish the peer review history of their article (what does this mean?). If published, this will include your full peer review and any attached files.

Reviewer #1: No

Reviewer #2: No
---

## [Editor Report · Decision Letter 1]

11 May 2023

Dear Dr. Rowley,

We are pleased to inform you that your manuscript '­­­Novel viruses of the family Partitiviridae discovered in Saccharomyces cerevisiae' has been provisionally accepted for publication in PLOS Pathogens.

Best regards,

Aiming Wang, Ph.D

Academic Editor

PLOS Pathogens

Shou-Wei Ding

Section Editor

PLOS Pathogens

Kasturi Haldar

Editor-in-Chief

PLOS Pathogens

orcid.org/0000-0001-5065-158X

Michael Malim

Editor-in-Chief

PLOS Pathogens

orcid.org/0000-0002-7699-2064
---

## [Editor Report · Acceptance letter]

5 Jun 2023

Dear Dr. Rowley,

We are delighted to inform you that your manuscript, "­­­Novel viruses of the family Partitiviridae discovered in Saccharomyces cerevisiae," has been formally accepted for publication in PLOS Pathogens.

Best regards,

Kasturi Haldar

Editor-in-Chief

PLOS Pathogens

orcid.org/0000-0001-5065-158X

Michael Malim

Editor-in-Chief

PLOS Pathogens

orcid.org/0000-0002-7699-2064